# Pharmacological Treatment for Terminal Agitation, Delirium and Anxiety in Frail Older Patients

**DOI:** 10.3390/geriatrics9020051

**Published:** 2024-04-18

**Authors:** Dine A. D. Jennes, Tim Biesbrouck, Maaike L. De Roo, Tinne Smets, Nele Van Den Noortgate

**Affiliations:** 1Department of Geriatric Medicine, Antwerp University Hospital, 2650 Edegem, Belgium; 2Department of Geriatric Medicine, University Hospitals Leuven, 3000 Leuven, Belgium; 3Department of Public Health and Primary Care, Gerontology and Geriatrics, KU Leuven, 3000 Leuven, Belgium; 4End-of-Life Care Research Group, Vrije Universiteit Brussel [VUB] and Ghent University, 1090 Brussels, Belgium; 5Department of Geriatric Medicine, Ghent University Hospital, 9000 Ghent, Belgium

**Keywords:** frailty, terminal care, delirium, confusion, anxiety, review

## Abstract

Context: Psychological distress symptoms in the last days of life often contribute to the overall symptom burden in frail older patients. Good symptom management practices are crucial to ensure high-quality end-of-life care in an aging population, though the best pharmacological approach to treat these psychological symptoms has yet to be established. Objectives: To identify current evidence-based and practice-based knowledge of pharmacological interventions for the treatment of agitation, delirium, and anxiety during the last days of life in frail older patients. Methods: A systematic, mixed methods review was performed through MEDLINE via PubMed and EMBASE from inception until February 2022 and updated through March 2023. National and international guideline databases and grey literature were searched for additional studies and guidelines. Results: Four quantitative studies, two non-randomized and two descriptive, were identified. No randomized controlled trials met inclusion criteria. No qualitative studies were withheld. The three consensus-based protocols that were found through citation searching and screening of grey literature did not meet the standards for inclusion. Haloperidol is recommended in consensus-based guidelines for delirium and is widely used, but high-quality evidence about its efficacy is missing. Better control of agitation or refractory delirium might be achieved with the addition of a benzodiazepine. There is no evidence available about the treatment of anxiety in the last days of life in frail older patients. Conclusions: This mixed methods review demonstrates the lack of good quality evidence that is needed to help clinicians with pharmacological treatment decisions when confronted with psychological symptoms in the last days of life in frail older patients. Population aging will only emphasize the need for further research in this specific population.

## 1. Introduction

Worldwide, the number of persons that are 80 years or older is expected to triple by 2050. As a result, population aging is becoming one of the biggest public and medical health challenges of this time [1].

Care of an older and frail population demands a customized approach because of changing homeostasis, presence of multimorbidity, altered pharmacokinetics, and an increased risk of developing atypical symptoms such as delirium [2]. The medical focus shifts from curation and survival to the preservation of functional ability and quality of life. At some point, this quest for quality turns into the challenge to ensure the right to a dignified death. Increased knowledge among health care professionals about palliative care and end-of-life care in an older population is necessary to allow for a good quality of dying.

When death is imminent, not only physical but also psychological symptoms, such as delirium, agitation, and anxiety, can arise. Increased frailty and cognitive decline may add difficulty to the recognition of these symptoms and the provision of adequate symptom control in these last days of life. 

According to DSM-V criteria, delirium is defined as an entity of (sub)acute onset of fluctuating attention and awareness with decreased performance in one or more cognitive domains and is related to an underlying medical illness, drug exposure, or drug withdrawal [3]. In the last days of life, the cause of delirium is often non-reversible and multifactorial, requiring a symptom-oriented rather than a curative approach.

Agitation can be seen in the mixed and hyperactive delirium subtypes but can also occur outside of the stringent definition of delirium [4]. Anxiety may be the consequence of other symptoms such as dyspnea, but it can also translate the fear of anticipated suffering or the fear of death itself. The presence of anxiety can exert a negative impact on the quality of dying and deserves appropriate attention and treatment if deemed necessary [5]. In the European PACE study, psychological distress symptoms in the last days of life of frail nursing home residents were seen in different numbers. Anxiety was reported in 55% of the dying residents in England and up to 77% in Poland [6]. The use of a pharmacological therapy, e.g., antipsychotics and sedatives, to treat these symptoms also varied significantly between the different countries in this study [7]. 

In a Dutch trial of 332 patients with dementia dying in nursing homes, the presence of agitation was associated with lower quality of life scores provided by nurses or physician. This finding indicates a level of perceived suffering by family and caregivers and advocates for proper treatment practices. In that same trial, 35% of all patients needed treatment for agitation in the last week of life. Anxiolytics were given to 57%, antipsychotics to 50%, and physical restraints were used in 5% of the agitated patients in this cohort [8]. 

In a cross-sectional descriptive study in a Belgian acute geriatric ward, there was significantly less deprescribing and less provision of anticipatory prescriptions prior to death in patients suffering from dementia. In their overall cohort, only 15% of all patients were given benzodiazepines [9]. 

To improve the management of psychological distress symptoms in the last days of life in older patients with frailty, this review aims to identify current evidence-based and practice-based knowledge of pharmacological interventions for the treatment of agitation, delirium, and anxiety. 

## 2. Methods

### 2.1. Design

A systematic mixed methods review was conducted (PROSPERO review protocol nr. CRD42022306178). Results were reported following the Preferred Reporting Items for Systematic Reviews and Meta-Analysis (PRISMA).

### 2.2. Search Strategy (Review Protocol: Appendix A, Search Protocol: Appendix B)

PUBMED/MEDLINE and EMBASE were searched from inception to 9 March 2023. National (EbPracticenet, pallialine.be), international guideline databases (NICE, G.I.N, EBM guidelines, SIGN, NHG, NVKG, pallialine.nl), and grey literature were explored for guidance concerning the treatment of psychological symptoms in the last days of life in a frail older population.

### 2.3. Eligibility Criteria

Publications were eligible for inclusion when they were written in English, Dutch, or French, when they reported results of quantitative or qualitative studies, and when full text was available. Experience-based protocols were included if they provided a clear methodology. Due to the expected limited number of articles focusing on the treatment of symptoms in the last days of life in older people, a broad definition for “older people” was used, including studies where the mean and median age was above or equal to 65 years and the lower limit of the range and interquartile range were above or equal to 55 years. Different indicators for the presence of terminal psychological symptoms were accepted for inclusion.

Studies in which patients were treated in the intensive care unit or psychiatric ward and/or studies concerning a perioperative setting were excluded. Non-pharmacological interventions and treatment of psychological symptoms in a palliative setting outside of the terminal phase were excluded.

### 2.4. Data Extraction and Data Analysis

Two reviewers screened the titles and abstracts independently, using the open access software Rayyan. Disagreements between reviewers were primarily resolved by open discussion, and by a third party in case of a persistent lack of consensus. Data extraction with special attention for population characteristics and frailty characteristics was conducted with quality control by a second reviewer. Frailty was assessed through frailty scores or any other descriptive sign of age-related vulnerability, as listed in the evidence tables. 

### 2.5. Quality Assessment

The two reviewers independently scored the quality of evidence using the Mixed Methods Assessment Tool (MMAT) version 2018.

### 2.6. Outcome

The primary objective was to determine the best pharmacological treatment to achieve adequate symptom control of agitation, delirium, and anxiety. Secondary outcomes that were considered, when available, were adverse side effects and the impact of the pharmacotherapy on life span.

## 3. Results

### 3.1. Study Selection (Prisma Flowchart, Appendix C)

Searches in PubMed/MEDLINE and EMBASE revealed 1345 results containing 1178 unique entries. Based on the screening of title and abstract, 1062 articles were excluded. Of the remaining 96 articles, 25 studies could not be retrieved for full-text evaluation, leaving 71 studies to be evaluated for eligibility. A total of 67 studies did not fulfil inclusion criteria, making 4 the total number of included studies.

Citation searching revealed 62 articles, of which 27 were sought for retrieval. Guideline databases disclosed 48 relevant guidelines and 1 website. None of these articles nor guidelines were found eligible after evaluation of full-text, content, and methodology.

Comparison and combined analysis of the results was impossible because of the limited number of studies lacking sound methodology and the heterogeneity between interventions and outcome measures.

### 3.2. Study Results

Of the four remaining studies, two were quantitative non-randomized studies and two were descriptive studies. No randomized controlled trials were eligible for inclusion (Table 1, Appendix A).

The two retrospective studies investigated delirium [10,11]. Delirium was defined using non-standardized descriptions in both studies. One study was conducted in a nursing home and the second one in a palliative care ward. Schildmann et al. [10] studied the use of sedatives with and without continuous effect in a multicenter retrospective cohort of predominantly poly-pathology patients in German nursing homes. Continuous effect was defined as a continuous parenteral administration or repeated administrations with similar effect. The study included 512 patients with a median age of 89 years. Sedatives were used in one fifth of patients, and 42% of sedatives had a continuous effect. The most often used sedative was lorazepam with a median daily dose of 1 mg (range 0.5–6 mg). The portion of patients receiving sedatives in the last week of life differed between nursing home facilities after correction for potential confounding factors. Patients receiving sedatives were younger (*p* < 0.001), had fewer diagnosis of dementia (*p* = 0.006) and were more often followed by a palliative care team consult [10].

The other retrospective study by Tatokoro et al. [11] compared age groups within a cohort of 1032 terminal cancer patients and showed a decreasing trend in the prevalence of pain, dyspnea, fatigue, and anxiety as the patients got older. The prevalence of delirium, however, remained the same in all age groups. The use of benzodiazepines decreased significantly with increasing age. In the age group under 70 years, 84.8% of people needed benzodiazepines compared to 69% in the age group of 90 years and over. Furthermore, there was a non-significant decreasing trend in the use of antipsychotics with older age but 30–35% of the oldest patients still received antipsychotics in the last days of life [11].

Of the two descriptive studies, one took place in a general hospital and one in a long term care facility for US veterans; one examined delirium and the other one investigated agitation and restlessness without a clear definition, respectively [12,13].

Gambles et al. [12] studied the use of medication for agitation and restlessness in patients whose care was supported by the UK’s ‘Liverpool care pathway’ for the dying patient. In this study, a retrospective chart review of medication use in the terminal phase of 3893 predominantly older patients (median age 81 years) was performed. The majority were non-cancer patients. Half of all patients were treated with medication for agitation and restlessness, either with rescue medication alone [PRN], continuous subcutaneous infusion only (CSCI), or a combination of both (CSCI + PRN). Midazolam was used more often (93%, 87%, and 98%, respectively) than haloperidol (4%, 16%, and 17%) and levomepromazine (3%, 12%, 14%). The median total dose of midazolam in the last 24 h was 2.5 mg (90% CI 2.5–10 mg) on a PRN base and 10 mg (90% CI 5–20 mg) with CSCI. For patients with a combination of PRN and CSCI, the median total dose in the last 24 h was higher (15 mg, 90% CI 7.5–40 mg). The median total day dose of haloperidol was 1.5 mg (90% CI: 0.5–2.85 mg) in the PRN group, 3 mg (90% CI: 1.5–5 mg) in the CSCI only group and 3 mg (90% CI: 1.5–10 mg) in the PRN and CSCI combination group. Median survival was 47 h in the CSCI only group and 27 h in the PRN only group. Information about effectiveness and safety was absent [12].

The usage pattern of pharmacological therapy at the end-of-life was a secondary descriptive outcome measure in a cohort study of 276 veterans with a mean age of 75 years that was conducted by Ellsworth et al. [13]. In the last two weeks of the veterans’ lives, 67.4% required antipsychotics. The most commonly used antipsychotic drug was oral or subcutaneous haloperidol (94% of the time). A small number of patients 4.4% (*n* = 4) used haloperidol for nausea and vomiting and not for delirium. The use of antipsychotic drugs was correlated with the use of steroids, opioids, and anticholinergics in this study [13].

**Table 1 geriatrics-09-00051-t001:** Evidence table.

Author	Symptom	Study Type	Population	Intervention	Outcome
Schildmann (2021) [10]	Agitation, anxiety, delirium/hallucinations without clear definition	Quantitative non-randomized trial Multicenter retrospective cohort study	Nursing homes 512 patients Age: 89 y (range 55–07) Majority with multi-pathology	Use of sedatives and use of sedatives with continuous effect	Primary outcome: use of sedatives generally in the last week of life and use of sedatives with continuous effect Secondary outcome: factors associated with the use of sedatives
Tatokoro (2022) [11]	Delirium, anxiety without a clear definition and other symptoms	Quantitative non-randomized trial Retrospective cohort study	In patient palliative care unit 1032 patients Age: median 79 y (IQR 71–86) All cancer patients; most common cancer being pancreatic cancer ECOG PG, palliative performance status Preterminal phase	Treatment of pain and dyspnea with opioids. Use of benzodiazepines, antipsychotics, anti-emetics, anticholinergics Groups: age classes (<70 y), (70–79 y), (80–89 y), (90 y or older)	Primary outcome: symptom prevalence Secondary outcome: need for opioids, need for sedation, need for benzodiazepines, need for antipsychotics
Ellsworth (2021) [13]	Delirium	Quantitative descriptive study Retrospective case-control study	Long term care hospice unit 276 patients Age: overall mean age 75.5 y All veterans Mostly cancer diagnosis, 24% and 18.9% dementia diagnosis in group 1 and 2 respectively Frailty characteristics not reported Evaluation two weeks prior to death	Group 1: use of antipsychotics in the last 2 weeks of life Group 2: no use of antipsychotics in the last 2 weeks of life	Primary outcome: determination of risk factors Secondary outcome: current usage patterns in treatment for terminal delirium
Gambles (2011) [12]	Agitation and restlessness without a clear definition	Quantitative descriptive study Retrospective epidemiological study	General hospitals 3893 patients Age: Median 81 y 66% diagnoses other than cancer Frailty characteristics absent Terminal phase	Recording use of medication for agitation and restlessness in the final days of life supported by the Liverpool Care pathway: midazolam, haloperidol, levomepromazine	Primary outcome: usage of medication PRN and CSCI for midazolam, haloperidol and levomepromazine Secondary outcome: life span

### 3.3. Quality of the Studies

The efficacy of individual drugs could not be established due to the lack of placebo-controlled trials with an adequate representation of our target population. There was a heterogeneity in outcome measures and studied populations within the scarce existing literature. A high risk of attrition was present in at least one study and not all studies were transparent concerning their confounders. 

## 4. Discussion

This mixed methods review synthetizes the available knowledge about the pharmacological treatment of psychological symptoms in the last days of life in an older population with frailty. Very little evidence about the treatment of agitation, delirium, and anxiety in this specific population was found. 

A decreasing trend of sedative use with increasing age and in nursing home residents with dementia near the end-of-life was seen in the last week of life in two retrospective studies [10,11]. Whether these patients truly needed less sedatives or whether their symptomatology was underappreciated is unknown due to missing information about comfort-related outcome measures in these publications.

Epidemiologic research indicates that the use of haloperidol in the last days of life is common [13]. However, in the absence of good quality placebo-controlled trials and in the presence of inconsistent results from real life data, the efficacy of haloperidol remains unestablished. This finding is in line with a Cochrane systematic review examining drug therapies for terminal delirium in a general adult population [14]. Anxiety was the least studied psychological distress symptom. No information was found concerning the treatment of anxiety apart from delirium or dyspnea, confirming the findings of an empty Cochrane review conducted by Salt et al. in a general adult population [15]. 

The predefined eligibility criteria led to exclusion of five RCT’s that studied cancer patients taken care of in an inpatient palliative care unit. The average age in these studies was below the defined cut-off criteria and age ranges were too wide to be considered for inclusion and to allow for proper representation of the intended geriatric target group. Nevertheless, a part of this population could be considered as older people with some frailty characteristics, yielding potential valuable information for the present study [16,17,18,19,20]. Most of these studies used haloperidol in one of the intervention arms. An unblinded randomized study of 79 patients with a wide age range but mean age around 65 years by Ferraz Gonçalves et al. [16] compared haloperidol plus midazolam subcutaneously with haloperidol in monotherapy. The results demonstrated a more effective (84% compared to 64% after the first dose) and faster (median time 15 min compared to 60 min) control of agitation with the combination therapy, but increased sedation was reported. Outcome measures were not standardized and the unblinded character of the study does increase the risk of reporting bias [16]. A small study with proper blinding and clearly described outcome measures by Hui et al. [17] examined 54 patients with a mean age of 62 years. In this trial, all patients were given open label haloperidol followed by closed label intravenously administered haloperidol 2 mg plus lorazepam 3 mg versus haloperidol 2 mg plus placebo. The combination group showed reduced agitation scores and decreased need for rescue neuroleptics, although delirium severity scores remained comparable after 24 h of treatment [17]. A pre-planned secondary analysis of the same study population by Tang et al. showed a significant difference in the need for rescue medication between the two groups after 8 h (15% in the combination group and 62% for haloperidol only) [18]. Lin et al. studied the use of oral olanzapine (mean age 61 years) compared to oral haloperidol (mean age 68 years) in 30 terminal and preterminal cancer patients. Most but not all patients were comfortable with a low dose (5 mg) of olanzapine and a low dose (5 mg) of haloperidol and no significant difference in response was seen between groups [19]. Hui et al. also conducted a parallel group-randomized trial with initial recruitment in open label haloperidol and secondary randomization of 45 cancer patients (ages between 55 and 75) in a ‘haloperidol escalation’ group, a ‘rotation to chlorpromazine’ group, or a ‘combination of haloperidol and chlorpromazine’ group for the treatment of refractory delirium. Patients with Parkinson’s disease and Alzheimer’s dementia were excluded from this study. Significant within-group reduction of the RASS (Richmond Agitation and Sedation Scale) was seen in all three groups without significant between-group differences. In the combination group, the need for rescue neuroleptics or benzodiazepines was higher but no differences were seen between groups in the ability to reduce agitation [20]. The patients in these RCT’s are not fully representative of the frail older patients in the scope of this review. The lack of evidence in this specific population is not surprising, since research on the last days of life is difficult and challenging because of the ethical considerations concerning the participation of a very vulnerable population [21]. These ethical barriers, in combination with the known underrepresentation of frail older people in pharmacological research in general, can explain the limited evidence that was found about the intended target population [22]. 

Grey literature did reveal some guidelines meeting the population requirements, but they all lacked methodological substantiation and should be considered solely consensus based. The Brisbane South Palliative Care Collaborative developed a toolkit for general practitioners in 2015 with a guide to the pharmacological management of end-of-life symptoms in residential aged care [23]. The Canadian Coalition for Seniors’ Mental Health adapted an evidence- and consensus-based guideline for the assessment and treatment of delirium in a palliative setting and the Dutch guideline for the treatment of delirium in frail and older patients added a section on the treatment of delirium in palliative care to its revision in 2020 [24,25]. The recommendations in this last guideline were largely based on a consensus-based guideline from the IKNL (Dutch Integrated Cancer Institute) with little transparency about methodology and the absence of evidence-level grading [26].

These three protocols from different parts of the world all recommend the use of low dose haloperidol as a primary choice for the treatment of delirium at the end-of-life. Starting doses between 0.5 mg and 1–2 mg twice daily are suggested. The Canadian guideline leaves room for the use of atypical antipsychotics (without further specification) to avoid extrapyramidal side effects and the Dutch guideline warns for an increased risk of extrapyramidal symptoms with haloperidol dosage above 4.5 mg/day. This warning seemed to be based on the results of one study with high attrition bias that was discussed in a Cochrane review from 2007, examining the use of antipsychotics outside of the palliative care setting [27,28]. The updated version of this Cochrane review did not establish the same effect after pooling of the results [29]. The rationale behind the recommended maximum dosages in the Dutch guideline of 10 mg haloperidol for parenteral administration and 20 mg for oral administration is not clearly stated. In current consensus-based guidelines, the use of benzodiazepines is recommended in case of refractory delirium and for the treatment of anxiety. 

### 4.1. Strengths and Limitations

This review applied an extensive search strategy in order to identify different types of evidence regarding the treatment of psychological symptoms in a terminal frail older population. Publications including RCT’s, non-randomized trials, real life epidemiological data, and consensus protocols used in clinical practice were all assessed for inclusion.

Limitations are the small number of searched databases, and the descriptive approach used to analyze data and assess quality. Registers were not examined. It is uncertain whether the studies for which the full text could not be retrieved predisposes to a selection bias. 

### 4.2. Conclusions and Recommendations

This systematic mixed-methods review found very limited evidence on the pharmacological treatment of the psychological distress symptoms delirium, agitation, and anxiety in dying frail older patients. Heterogeneity in symptom definitions and outcome measures in the existing research complicate the interpretation, comparison, and integration of results. 

This calls for increased efforts to conduct research in the population of older patients with frailty. A joint research agenda with uniform symptom definitions adapted to the palliative care setting and comparable outcome measures should be aspired by the research community. Thinking out of the box to produce study designs that conquer ethical barriers without compromising the quality of care in the last days of life for frail patients and their caregivers will be the challenge for future research.

We confirm that neither the manuscript nor any parts of its content are currently under consideration or published in another journal. All authors have approved the manuscript and agree with its submission to Geriatrics.

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
