# Peer review of "Pharmacological Treatment for Terminal Agitation, Delirium and Anxiety in Frail Older Patients"

_geriatrics, 2024, doi:10.3390/geriatrics9020051_

Round 1
Reviewer 1 Report
Comments and Suggestions for Authors
Thank you for the opportunity to review this paper which reports a systematic review which aimed to address the questions on evidence and practice knowledge on pharmacological interventions for the treatment of agitation, delirium and anxiety during the last days of life in frail older patients. This appears a well structured review although as authors state searching more data bases may have elicited other papers. I wonder if the authors consider the context of care i.e the difference between nursing home and specialist palliative care facility influences medication regimes. It seem that with the ethical issues of use of placebo drugs further research might explore clinicians decision making to understand if qualitative judgements are made about type of symptom and need influenced by age.
Author Response
Comment: I wonder if the authors consider the context of care i.e the difference between nursing home and specialist palliative care facility influences medication regimes.
Response:
This review was conducted to support the development of a terminal care guideline/ protocol to be used in the context of a nursing home. However, because of the anticipated low number of good quality research in this specific setting, early on, the decision was made to expand the setting in de review protocol to other settings. Postoperative and intensive care environments were excluded because treatment in these resource rich environments, would not be easily applicable in resource poor environments. The authors considered that especially the routes of administration and the monitoring would differ between settings rather than the chosen therapy or dosages. Availability of drugs might differ more between countries than between mentioned settings within a country.
Reviewer 2 Report
Comments and Suggestions for Authors
the manuscript report a systmeatic review that investigate pharmacological treatments for old patients in terminal palliative care with agitation, delrium and anxiety.
Methods are clearly explained and are relevant.
Resultas shown the paucity of tThe manuscript reports a systematic review investigating pharmacological treatments for older terminal palliative care patients with agitation, delirium and anxiety.
Methods are clearly explained and relevant.
The results show the paucity of studies and no clear effects were documented in the four studies included in the review.
The conclusion is relevant. No clear effect was docuùented in the four studies included in the review.
Author Response
Thank you for the review. No additional adaptations seem to have been asked by this reviewer.
Reviewer 3 Report
Comments and Suggestions for Authors
It's a very interesting review, my congratulations.
Just one minor issue.
Introduction
Line 42 “Population aging will therefore become one of the biggest public and medical health challenges of this time”
Line 49 “Increased knowledge among health care professionals about palliative care and end-of-life care in an older population will be necessary to allow for a good quality of dying”
I totally agree on your perspective regarding the importance of palliative care. From my clinical experience, advancements in medical science alongside cultural factors have notably heightened concerns around tanatophobia, particularly among caregivers (https://doi.org/10.3390/healthcare12010107) and physicians (https://doi.org/10.2190/TFG3-YQJQ-E0M6-4). This trend poses a significant risk, manifesting in a psychological aversion to the prospect of patient death. Consequently, it may lead to the unnecessary prolongation aggressive life-prolonging treatments that are ultimately futile. This issue demands immediate attention, not future consideration. Therefore, I suggest shifting from the future tense to the present continuous tense to reflect the ongoing nature of this challenge.
Comments on the Quality of English LanguageThe manuscript requires minor english editing and formatting according to the journal guidelines.
Author Response
Thank you for the review and the suggestions. We hope that the following adaptations correctly capture the remarks.
|
Reviewer 3: Line 42 “Population aging will therefore become one of the biggest public and medical health challenges of this time”
|
As a result, population aging is turning into one of the biggest public and medical health challenges of this time. |
|
Reviewer 3: Line 49 “Increased knowledge among health care professionals about palliative care and end-of-life care in an older population will be necessary to allow for a good quality of dying” I totally agree on your perspective regarding the importance of palliative care. From my clinical experience, advancements in medical science alongside cultural factors have notably heightened concerns around tanatophobia, particularly among caregivers (https://doi.org/10.3390/healthcare12010107) and physicians (https://doi.org/10.2190/TFG3-YQJQ-E0M6-4). This trend poses a significant risk, manifesting in a psychological aversion to the prospect of patient death. Consequently, it may lead to the unnecessary prolongation aggressive life-prolonging treatments that are ultimately futile. This issue demands immediate attention, not future consideration. Therefore, I suggest shifting from the future tense to the present continuous tense to reflect the ongoing nature of this challenge.
|
Increased knowledge among health care professionals about palliative care and end-of-life care in an older population is becoming necessary to allow for a good quality of dying
|